# Association between Colorectal Adenoma and Carotid Atherosclerosis in Korean Adults

**DOI:** 10.3390/ijerph15122762

**Published:** 2018-12-06

**Authors:** Hyunji Kim, Yoon Jeong Cho, Yun A. Kim, Sang Gyu Gwak

**Affiliations:** 1Department of Family Medicine, School of Medicine, Daegu Catholic University, Daegu 42472, Korea; 170062@dcmc.co.kr (H.K.); yuna815@dcmc.co.kr (Y.A.K.); 2Department of Medical Statistics, School of Medicine, Daegu Catholic University, Daegu 42472, Korea; sgkwak@cu.ac.kr

**Keywords:** colorectal adenoma, carotid plaque, intima-media thickness, metabolic syndrome, atherosclerosis

## Abstract

*Background:* Colorectal neoplasm, including colorectal adenoma, is associated with old age, cigarette smoking, and the presence of metabolic syndromes. These are also risk factors for cardiovascular disease. Carotid ultrasonography is a noninvasive test that can predict the risk of cardiovascular disease and may be another test that may provide indications of these risk factors. This study aimed to investigate the association between colorectal adenomatous polyps and carotid atherosclerosis. *Methods:* This study included 548 adults who underwent colonoscopy and carotid ultrasonography for a health examination between March 2013 and December 2017 at a university hospital in South Korea. Abnormal carotid sonography findings included either increased carotid intima-media thickness or presence of carotid plaques. *Results:* The proportion of subjects with overall colorectal adenomatous polyps was 31.0% (170/548). Colorectal adenoma was more prevalent in the presence of abnormal carotid ultrasonography findings (38.6% vs. 27.6%, *p* = 0.013). Colorectal adenomatous polyp was significantly associated with abnormal carotid ultrasonography findings (OR = 1.65; 95% CI, 1.12–2.42, *p* = 0.011) in a multivariate analysis after adjusting for age, sex, cigarette smoking, alcohol consumption, and presence of metabolic syndrome. *Conclusion:* Colorectal adenoma is significantly associated with carotid atherosclerosis.

## 1. Introduction

Colorectal cancer (CRC) is the third-most commonly diagnosed cancer in both men and women and the second leading cause of cancer-related death worldwide [1]. Colorectal adenoma is a premalignant lesion, and removal of these adenomas can reduce mortality rates associated with CRC [2]. It is important to identify individuals who have a high risk for CRC. Interestingly, cardiovascular disease (CVD) shares common risk factors with CRC, such as old age, cigarette smoking, sedentary lifestyle, high-fat diet, and presence of metabolic syndrome. Recently, many studies have reported the significant association between CVD and CRC [3,4,5,6,7]. Carotid artery disease has been associated with CVD and an increased risk of cardiovascular (CV) events [8,9]. Carotid artery sonography has been commonly used as a noninvasive test that can predict the risk of CVD and may be another test for the identification of these risk factors. Intima-media thickness (IMT) of the internal carotid artery (ICA) reportedly adds value to the Framingham risk score [10]. Measuring carotid IMT (CIMT) has been shown to predict CV risk in multiple large studies. Carotid plaque is a more powerful predictor of CV risk compared with CIMT alone [11]. Furthermore, adding a thick CIMT to carotid plaque increases the prognostic power for cardiac events [12]. The aim of this study was to analyze the association between carotid atherosclerosis and colorectal adenoma as a risk factor for CRC based on screening colonoscopy and carotid artery sonography findings.

## 2. Materials and Methods

### 2.1. Study Population

This study included 564 adults who underwent a screening colonoscopy and carotid ultrasonography on the same day for a routine health check-up from March 2013 to December 2017 in the health promotion center of Daegu Catholic University Hospital in South Korea. Sixteen subjects who underwent the test twice during this period were excluded. A total of 548 subjects (412 men and 136 women) aged between 29 and 80 years were analyzed. This study was approved by the Institutional Review Board of the Daegu Catholic University Hospital (CR-18-057).

### 2.2. Methods

#### 2.2.1. General Characteristics and Lifestyle Factors

Participants were interviewed by a trained interviewer about their demographic characteristics and lifestyle using a standardized questionnaire. Anthropometric measurements and a blood test were conducted after 12 h of fasting. The waist circumference was measured at the midpoint between the lower margin of the rib cage and upper margin of the iliac crest during minimal respiration. Obesity was defined as having a body mass index (BMI) of ≥25 kg/m^2^. 

#### 2.2.2. Colonoscopy

We performed colonoscopy into the cecum after bowel preparations using a 2 L polyethylene glycol lavage. All colonoscopies were performed by board-certified gastroenterologists. All polyps were biopsied or resected, and a histopathological assessment was performed. An adenoma was defined as a polyp with adenomatous tissue, including tubular, sessile, and serrated adenoma and advanced lesion, as confirmed by pathology. The non-adenomatous polyp group includes other polyps like inflammatory polyps, hyperplastic polyps, etc., as well as no polyps. The characteristics, including polyp size, number, location, pathologic findings, and the presence of advanced lesions were noted. The size of a polyp was assessed by comparing to an open biopsy forceps. The largest size was used in the case of multiple polyps. An advanced lesion of the colon was defined as either a size ≥1 cm, ≥3 polyps, or containing villous features and/or high-grade dysplasia in pathologic findings.

#### 2.2.3. Carotid Sonography

Carotid plaques and CIMT were assessed using B-mode ultrasonography. CIMT was measured in the common carotid artery (CCA). CIMT measurements were taken at the point of maximal thickness in the walls of the carotid arteries, using a measurement tool in the ultrasound scanner. Increased IMT was defined as an IMT ≥ 1.0 mm. The CCA, ICA, external carotid artery, and carotid bulb were examined, and the presence of atherosclerotic plaques in the longitudinal and transverse planes was recorded. The presence of atherosclerotic plaques was defined as local lesions with a protrusion into the arterial lumen. Abnormal carotid sonography findings as carotid atherosclerosis included either increased CIMT or the presence of carotid plaques.

#### 2.2.4. Assessment of Risk Factors

Current smoking was defined as smoking at least one cigarette per day for the previous 12 months. A metabolic syndrome was defined according to the criteria by the International Diabetes Federation (IDF) with a modified waist circumference appropriate for Koreans. According to these criteria, individuals were defined as having an metabolic syndrome if three or more of the following clinical features were present: (1) central obesity with WC ≥90 cm for men, ≥85 cm for women (2) triglyceride ≥150 mg/dL, (3) high-density lipoprotein cholesterol ≤40 mg/dL for men or ≤60 mg/dL for women, (4) elevated blood pressure ≥130/85 mmHg or use of antihypertensive agents; and (5) fasting plasma glucose ≥100 mg/dL or HbA1c ≥6.4% or use of diabetic medications [13]. High risk alcohol consumption was defined as an average drinking rate of more than seven (more than five for women) glasses per week and more than twice a week [14].

### 2.3. Statistical Analysis

General and colonoscopic characteristics were summarized using descriptive analysis. The values were presented as mean ± standard deviation (SD) for quantitative variables and frequency (percentage) for qualitative variables. Univariate analysis was performed for the comparison of characteristics of colorectal adenomatous polyp and carotid atherosclerosis using a two sample *t*-test for quantitative variables and a chi-square test for qualitative variables. In multivariate analysis, binary logistic regression analysis was performed to examine the association between carotid atherosclerosis and colorectal adenoma after adjustment for age, sex, cigarette smoking, alcohol consumption, and the presence of a metabolic syndrome and its components. The interaction effect was also checked and the odds ratio (OR), 95% confidence interval (95% CI), and *p*-value are presented. Data analysis was performed by a medical statistician. All tests were two-sided, and a *p*-value < 0.05 indicated statistical significance. IBM SPSS version 19.0 (IBM, Armonk, NY, USA) was used in the analysis.

## 3. Results 

The mean age of the study subjects (mean ± SD) was 54.96 ± 8.0 years in the adenoma group and 52.2 ± 8.5 years in the non-adenoma group (*p* = 0.00). The prevalence of adenoma was higher in men (82.9%) than women (17.1%). Colorectal adenoma was significantly associated with a higher mean BMI, waist circumference, and triglyceride, fasting glucose, and glycated hemoglobin (HbA1c) levels, and a lower mean high-density lipoprotein (HDL) cholesterol level. The prevalence of colorectal adenoma was significantly higher in subjects who currently smoke and consume heavy alcohol, and with diabetes, metabolic syndrome, and carotid atherosclerosis (*p* < 0.05) (Table 1). 

A comparison between the subjects with carotid atherosclerosis and normal carotid showed that colorectal adenoma was more prevalent in the carotid atherosclerosis group (38.6% vs. 27.6%, *p* = 0.01). However, there was no difference in location, size, and number of adenomatous polyps between the two groups. Moreover, there was no association between carotid atherosclerosis and a villous feature or high-grade dysplasia and advanced lesions (Table 2).

The risk of developing adenomatous polyps was divided by the abnormal carotid findings. After adjusting for covariates (age, sex, cigarette smoking, alcohol consumption, and presence of metabolic syndrome), the OR (95% CI) was 1.65 (1.12–2.42) for the colorectal adenomatous group (Table 3). Furthermore, increased CIMT and the presence of carotid plaque were also an independent risk factor for colorectal adenoma after adjusting for covariates. We analyzed the interaction between sex and carotid atherosclerosis. According to the result, there was no statistically significant interaction between sex and carotid atherosclerosis (OR: 0.972, 95% CI: 0.335 to 2.819, *p*-value: 0.959).

## 4. Discussion

We found that colorectal adenoma was significantly associated with carotid atherosclerosis in Korean adults after adjusting for covariates. In previous studies, Lee et al. [15] showed that coronary heart disease increases the risk for colorectal neoplasm using the Framingham Risk Score/Adult Treatment Panel III. Other studies have shown the relationship between atherosclerotic disease and colorectal neoplasm using the coronary calcification score and coronary diagnostic test, such as a coronary angiogram, to reflect atherosclerosis [16,17,18]. A meta-analysis also shows a positive relationship between ischemic heart disease and colorectal neoplasm [4]. Moreover, the findings of our study are not markedly different from those of previous studies reporting that atherosclerotic disease is positively correlated with colorectal neoplasm. Among these previous studies, there were few studies on carotid artery sonography. Carotid sonography, in which the IMT and presence of plaque or stenosis of the walls of the carotid artery are measured, is a noninvasive and comfortable test that can predict the risk of atherosclerotic disease. In the previous studies, CIMT is widely studied as a surrogate marker for detecting subclinical atherosclerosis for risk prediction and disease progress to guide medical intervention. The study of sensitivity and specificity of carotid ultrasound was conducted. The results were shown that the threshold of peak systolic velocity ≥130 cm/s is associated with a sensitivity of 98% (95% confidence intervals [CI], 97% to 100%) and specificity of 88% (95% CI, 76% to 100%) in the identification of angiographic stenosis of ≥50%. For the diagnosis of angiographic stenosis of ≥70%, a peak systolic velocity ≥200 cm/s has a sensitivity of 90% (95% CI, 84% to 94%) and a specificity of 94% (95% CI, 88% to 97%) [19]. Stein et al. [20] revealed that carotid sonography can be a good imaging method to detect the early stages of atherosclerosis and improve the risk stratification of patients. However, studies on the association between CIMT or carotid plaque on carotid artery sonography and colorectal neoplasm are rare. The reason is that the assessment of CIMT is not standardized. Evaluation of CIMT varies depending on the comprehensiveness of CIMT assessment: the number of carotid segments (CCA, ICA, or the carotid bulb), type of measurement made (mean or maximum of single or multiple measurements), and whether plaques were included in the IMT measurement. These methodological discrepancies limit the interpretation of results and have caused confusion on the role of IMT in CV risk prediction [21]. CIMT alone has weak predictive value for CVD while CIMT, including plaque presence, consistently improves the predictive power. Inclusion of plaque to the CIMT measurement has consistently been shown to improve the predictive power for CVD and coronary events [12,22,23,24]. Therefore, we used either increased CIMT of CCA and presence of carotid plaques at the carotid segments to assess carotid atherosclerosis.

In this study, carotid atherosclerosis, abdominal obesity, high triglyceride and fasting glucose levels, and low HDL-cholesterol levels showed significant association with colorectal adenoma. This shows that the metabolic syndrome is associated with colorectal neoplasm. It is known that colorectal neoplasm is associated with male sex, obesity, diabetes, smoking, and metabolic syndrome [25,26]. Many studies reported an association between atherosclerotic disease and the presence of colorectal neoplasm because these two diseases share common risk factors. Central obesity and components of metabolic syndrome can result in visceral fat deposition that is associated with insulin resistance and high IGF1 levels, which can influence the carcinogenic processes [27]. Furthermore, another mechanism is a possible shared pathogenic factor, chronic inflammation [28]. Several epidemiologic studies observed positive associations between C-reactive protein and CRC risk [29]. In an observational study of Taiwan, it was shown that hyperglycemia with *H. pylori* infection increased the risk for synchronous colorectal adenoma and carotid artery plaque formation. The study suggested the endotoxin would cause atherosclerosis and colon neoplasm via the mechanism underlying chronic inflammation and the change of gut permeability [30]. Pro-inflammatory cytokines play a role in atherogenesis and the early stage of colorectal carcinogenesis [31]. This offers further evidence that inflammation plays a role in colorectal carcinogenesis. Inflammation is a potentially modifiable risk factor; therefore, aspirin and other non-steroidal anti-inflammatory drugs (NSAIDs) that decrease inflammation may reduce CRC and CV risks [32].

Several studies have shown that atherosclerosis was significantly associated with size, presence of high-grade characteristics, advanced lesions, and multiplicity of colorectal adenomas [33]. However, there was no significant association between these factors in our study. This may be because the sample size was small and not representative of the complete diseased population in Korea or the world, as this study was conducted at a single center. Thus, the results of this study were different from those of previous large-scale studies.

The present study has some limitations. First, the enrolled population was a health screening cohort. Therefore, the study population may not have been representative of the general population in Korea. Second, this was a cross-sectional study; therefore, it may not be possible to establish a causal relationship between carotid atherosclerosis and colorectal adenoma. Moreover, we did not assess insulin resistance markers, such as homeostasis model assessment of insulin resistance, and inflammatory markers (MIF, TNF-α, and IL-6). Finally, we did not consider the effects of all potential confounding variables, such as family history of CRC, personal history of aspirin or NSAID use, or dietary variables. These variables may also play an important role in the risk for colorectal adenoma and atherosclerosis. 

Our study was conducted to identify the factors indicative of atherosclerosis using carotid sonography findings including either increased CIMT or the presence of carotid plaques. Considering that most studies use cardiac computed tomography, carotid artery sonography is a simple and noninvasive method that can predict the risk of cerebrovascular disease. 

Since colorectal adenoma and atherosclerosis have common risk factors and pathogenesis, patients who are at high risk for developing coronary heart disease were found to have an increased risk for the overall presence of colorectal neoplasm. According to the results of our study and those of previous studies, regular colonoscopies can be recommended for patients who are at high risk for atherosclerosis.

## 5. Conclusions

In conclusion, among subjects who underwent a carotid sonography and colonoscopy in the single center, the prevalence of colorectal adenoma was higher in those with increased CIMT and the presence of carotid plaque. We found that carotid atherosclerosis can be an independent risk factor for colorectal adenoma after adjusting for age, sex, alcohol consumption, current smoking, and the presence of a metabolic syndrome.

## Reference

## Figures and Tables

**Table 1 ijerph-15-02762-t001:** General characteristics of subjects according to colorectal adenomatous polyp.

Variables	Adenomatous Polyp	Non-Adenomatous Polyp ^a^	*p*-Value
	(*n* = 170)	(*n* = 378)	
Age (years), (mean ± SD)	54.9 ± 8.0	52.1 ± 8.5	<0.001
Men, *n (%)*	141 (82.9%)	271 (71.7%)	0.050
Current smoking, *n (%)*	39 (22.9%)	85 (22.5%)	0.027
Heavy alcohol, *n (%)*	52 (30.6%)	81 (21.4%)	0.021
Diabetes, *n (%)*	25 (14.7%)	29 (7.7%)	0.011
Hypertension, *n (%)*	45 (26.5%)	88 (23.3%)	0.420
Dyslipidemia, *n (%)*	51 (30.0%)	66 (17.5%)	0.001
Metabolic syndrome, *n (%)*	50 (29.4%)	70 (18.5%)	0.004
Systolic BP (mmHg), (mean ± SD)	123.4 ± 11.6	121.4 ± 11.5	0.054
Diastolic BP (mmHg), (mean ± SD)	77.4 ± 8.3	75.8 ± 9.2	0.048
BMI (kg/m^2^), (mean ± SD)	24.9 ± 3.1	24.3 ± 3.1	0.036
Waist circumference (cm), (mean ± SD)	86.1 ± 8.4	83.9 ± 8.9	0.006
Total cholesterol (mg/dL), (mean ± SD)	189.1 ± 36.5	188.6 ± 36.4	0.874
Triglyceride (mg/dL)), (mean ± SD)	120.2 ± 74.4	103.4 ± 69.6	0.011
HDL-C (mg/dL), (mean ± SD)	52.3 ± 13.2	56.6 ± 16.8	0.003
LDL-C (mg/dL), (mean ± SD)	129.2 ± 34.8	126.2 ± 33.2	0.339
Fasting glucose (mg/dL), (mean ± SD)	104.1 ± 29.0	95.0 ± 16.8	0.000
HbA1c (%), (mean ± SD)	5.8 ± 1.1	5.6 ± 0.6	0.001
Increased CIMT, *n (%)*	39 (22.9%)	63 (16.7%)	0.053
Carotid plaque, *n (%)*	29 (17.1%)	45 (11.9%)	0.069
Carotid atherosclerosis, *n (%)*	66 (38.8%)	105 (27.8%)	0.007

BP, blood pressure; BMI, body mass index; HDL-C, high density lipoprotein cholesterol; LDL-C, low density lipoprotein cholesterol; HbA1c, hemoglobin A1c; CIMT, carotid intima-media thickness. ^a^ Other polyp like inflammatory polyp, hyperplastic polyp, etc., and no polyp.

**Table 2 ijerph-15-02762-t002:** Characteristics of colorectal adenomatous polyps according to carotid atherosclerosis.

Variables	Carotid Atherosclerosis (*n* = 171)	Normal (*n* = 377)	*p*-Value
Adenomatous polyp	66 (38.6)	104 (27.6)	0.010
Location			
proximal	38 (22.2)	77 (20.4)	0.632
distal	33 (19.3)	51 (13.5)	0.082
both	7 (4.1)	24 (6.4)	0.286
Size, mm			0.004
<5	148 (86.5)	341 (90.5)	
5 ≤ 10	21 (12.3)	21 (5.3)	
≥10	2 (1.2)	16 (4.2)	
Number			0.172
<3	159 (93)	361 (95.8)	
≥3	12 (7.0)	16 (4.2)	
Villous feature or high-grade dysplasia			0.900
yes	2 (1.2)	4 (1.1)	
no	169 (98.8)	373 (98.9)	
Advanced lesion			0.766
yes	13 (7.6)	26 (6.9)	
no	158 (92.4)	351 (93.1)	

Values are presented as number (%).

**Table 3 ijerph-15-02762-t003:** Adjusted odds ratios (95% confidence intervals) for colorectal adenomatous polyps using multivariate analysis.

Variables	Colorectal Adenomatous Polyps	*p*-Value
Age	0.99 (0.99–1.00)	0.001 *
Men (ref: female)	1.23 (0.69–2.18)	0.485
Smoking (ref: non-smoking)	1.25 (0.74–2.14)	0.406
Alcohol	1.48 (0.95–2.23)	0.082
Metabolic syndrome	1.80 (1.17–2.74)	0.007 *
Increased CIMT	1.58 (1.02–2.47)	0.041
Carotid plaque	1.70 (1.04–2.76)	0.034
Carotid atherosclerosis	1.65 (1.12–2.43)	0.011 *

CIMT: carotid intima-media thickness. *: Statistically significant with *p* < 0.05.

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
