# Peer review of "Association between Colorectal Adenoma and Carotid Atherosclerosis in Korean Adults"

_ijerph, 2018, doi:10.3390/ijerph15122762_

Reviewer 1 Report

This study assessed the association between colorectal adenoma and carotid atherosclerosis in Korean adults. Indeed, there are several overlapping risk factors for these outcomes.

Introduction:

Please update the first reference which is fairly old and newer version are available.

Line 35-36: The reference is not recent and there is only one however you suggest many studies. Please update with newer references.

Methods:

Line 61: change "until" to "into"

How was metabolic syndrome defined?

How was alcohol consumption defined?

Was physical activity captured at all?

It seems you want to compare plaques vs. CIMT but there is no stratification in your analyses only abnormal carotid sonography. Please break it down by type and possible do a sensitivity analysis looking at one versus the other.

Line 89 should read "p-value<0.05 indicated statistical significance", not >0.05.

Results:

Table 1 headings are confusing. Non-adenomatous polyp sounds like they had some other kind of polyp. Please reword to indicate no polyp.

How was hypertension defined?

How did you select covariates for your models?

Please indicate how many had CIMT or presence of carotid plaques.

Did you evaluate statistical interaction between sex and carotid atherosclerosis? Even though the main effects were not significant, there may be effect modification here by sex.

Although numbers are getting small in the cells, did you evaluate advanced lesions for those that had polyps?

Discussion:

Line 136: You state carotid plaque is a more powerful predictor of CV risk compared to CIMT alone. Please display the proportions of each in your data.

Line 145 suggests sex is an effect modifier. Please test for statistical interaction in your data.

Were other potential confounders not collected or just not considered in the models?

Conclusions:

Line 181: please add language to the conclusion indicating its relevance to those in this particular study from your center in South Korea as you indicate the study population may have not been representative of the general population.

Author Response

1. Introduction:

1) Please update the first reference which is fairly old and newer version are available.

Answer) Thank you for the comments. We revised in the document

2) Line 35-36: The reference is not recent and there is only one however you suggest many studies. Please update with newer references.

Answer) Thank you for the comments. We revised in the document

2. Methods:

1) Line 61: change "until" to "into"

Answer) Thank you for the comments. We revised in the document

2) How was metabolic syndrome defined?

Answer) We have added the comments for the article and added them as references.

 In this study, metabolic syndrome(MS) was defined according to the criteria by the International Diabetes Federation (IDF) with a modified WC appropriate for Koreans. According to these criteria, individuals were defined as having MS if three or more of the following clinical features were present: (1) central obesity with WC ≥90 cm for men, ≥2125px for women (2) triglyceride ≥150mg/dl (3) high-density lipoprotein cholesterol ≤40mg/dl for men and ≤60mg/dl for women (4) elevated blood pressure ≥130/85mmHg or use of antihypertensive agents; and (5) fasting plasma glucose ≥100mg/dl or HbA1c ≥6.4% or use of diabetic medications.

3) How was alcohol consumption defined?

Answer) We have added the comments for the article and added them as references.

High risk alcohol consumption was defined as the average drinking rate is seven (Five for women) or more glasses per week and more than twice a week dringking.

4) Was physical activity captured at all?

Answer) We investigated physical activity of subject and classified by intensity of the physical activity. (low, moderate, high) High activity was defined as 1) vigorous-intensity activity for more than 20 minutes over three days per week, consuming more than 1,500 metabolic equivalent task (MET)-minutes per week or 2) walking, moderate activity, or vigorous activity over 7 days per week, consuming more than 3,000 MET-minutes per week. Moderate activity was defined as: 1) more than 20 minutes of vigorous-intensity activity over three days per week; 2) at least 30 minutes of moderate activity or walking for more than five days per week; or 3) walking, moderate activity, and vigorous activity for more than 5 days per week and consuming more than 600 MET-minutes.

We didn’t consider as a covariate because the uni-variate analysis of physical activity showed no statistically significant result.

5) It seems you want to compare plaques vs. CIMT but there is no stratification in your analyses only abnormal carotid sonography. Please break it down by type and possible do a sensitivity analysis looking at one versus the other.

Answer) Thank you for the comments. But this study is not aimed to compare plaques vs CIMT. As meaning of carotid atherosclerosis was abnormal carotid sonography findings which include either increased CIMT or presence of carotid plaques. The combination of the two finding was more meaningful to reflect atherosclerosis.

6) Line 89 should read "p-value<0.05 indicated statistical significance", not >0.05. revised in the document

Answer) Thank you for the comments. We revised in the document

3. Results:

1) Table 1 headings are confusing. Non-adenomatous polyp sounds like they had some other kind of polyp. Please reword to indicate no polyp.

Answer) Thank you for the comments.

 An adenomatous polyp is defined as a polyp with adenomatous tissue, as confirmed by pathology. Non-adenomatous polyp includes the results of inflammatory polyp, hyperplastic polyp, etc. and no polyp. “Non-adenomatous polyp” doesn’t mean “no polyp”.

2) How was hypertension defined?

Answer) Hypertension was defined as a systolic blood pressure of ≥140 mmHg, a diastolic blood pressure of ≥90 mmHg and/or the current use of antihypertensive agents.

3) How did you select covariates for your models?

Answer) Covariates were selected after univariate analysis by polyp among variable. Among them, sex, age, smoker, high risk dringking and metabolic syndrome showed statistical significance.

4) Please indicate how many had CIMT or presence of carotid plaques.

Answer) It is shown in Table 2. The number of carotid atherosclerosis (either increased CIMT and/or presence of plaque) was 171/548 (31%). It is not written in the paper, but the number of only plaque of carotid sonograpy was 74/548 (13%).

5) Did you evaluate statistical interaction between sex and carotid atherosclerosis? Even though the main effects were not significant, there may be effect modification here by sex.

Answer) Thank you for the comments.

We analyzed the the interaction between sex and carotid atherosclerosis. According to the result, there was no statistically significant interaction between sex and carotid atherosclerosis. (OR: 0.972, 95% CI: 0.335 to 2.819, p-value: 0.959)

6) Although numbers are getting small in the cells, did you evaluate advanced lesions for those that had polyps?

Answer) The advanced lesions according to carotid atherosclerosis is shown in Table 2. We couldn’t find significant finding of advanced lesion in our study.

4. Discussion:

1) Line 136: You state carotid plaque is a more powerful predictor of CV risk compared to CIMT alone. Please display the proportions of each in your data.

Answer) We made a slight change of the article in this part. As meaning of carotid atherosclerosis, abnormal carotid sonography findings which include either increased CIMT or presence of carotid plaques, the combination of the two findings was meaningful. It is shown in Table 2.

2) Line 145 suggests sex is an effect modifier. Please test for statistical interaction in your data.

Answer) According to our analysis, there was no statistically significant interaction between sex and carotid atherosclerosis.  (OR: 0.972, 95% CI: 0.335 to 2.819, p-value: 0.959)

3) Were other potential confounders not collected or just not considered in the models?

Answer) Thank you for the comments.
We could not consider potential confounders like family history of CRC and personal history of aspirin or NSAID use and dietary variables. Because, the data had to rely on the questionnaire of the subjects., may be less objective. It is also written in the limitation.

5. Conclusions:

1) Line 181: please add language to the conclusion indicating its relevance to those in this particular study from your center in South Korea as you indicate the study population may have not been representative of the general population.

Answer) Thank you for your comment. We revised in the document

Reviewer 2 Report

In this manuscript the authors reported the results on the association between colorectal adenoma and carotid atherosclerosis in Korean adults. 548 patients is a significant number. The study was conducted with methodological rigor and originality. The interesting results obtained can be considered a starting point for further studies on the subject. The results of this study suggest that a colonoscopy is performed in all patients whose sonographic examinations show abnormal carotid findings included either increased carotid intima-media thickness or presence of carotid plaques. For these reasons the manuscript can be considered for publication.

Author Response

Answer) Thank you for your comment.

Reviewer 3 Report

This is an interesting epidemiological cross-over study on the association between colorectal adenoma and carotid atherosclerosis. The study was well conducted, the results are clear and properly discussed. Only one suggestion as follow:

The introduction/discussion could be improved by adding a recently published paper on Oncotarget 2017 Oct 26; 8(65).

Author Response

This is an interesting epidemiological cross-over study on the association between colorectal adenoma and carotid atherosclerosis. The study was well conducted, the results are clear and properly discussed. Only one suggestion as follow:

The introduction/discussion could be improved by adding a recently published paper on Oncotarget 2017 Oct 26; 8(65).

Answer) We inserted as a reference in the discussion. Thank you for your comment.

Reviewer 4 Report

The authors admit that this study has limitations in that the enrolled population was a health screening cohort. Therefore, as they state, the study population may not have been representative of the general population in Korea. It was also based upon those screened in a single institution.  The authors should therefore provide more information on how subjects were recruited for screening, whether the institution was the only one in a defined area, and if there were subjects approached for screening who declined to be screened, what their features were in comparison to those screened.

I am unfamiliar with carotid sonography being used as a screening test. On what basis was this introduced? Has it been evaluated in terms of reducing the morbidity and mortality from cerebrovascular disease?  What is its sensitivity and specificity when used as a screening test?

Author Response

1)     The authors admit that this study has limitations in that the enrolled population was a health screening cohort. Therefore, as they state, the study population may not have been representative of the general population in Korea. It was also based upon those screened in a single institution.  The authors should therefore provide more information on how subjects were recruited for screening, whether the institution was the only one in a defined area, and if there were subjects approached for screening who declined to be screened, what their features were in comparison to those screened.

Answer) Thank you for your comments.

 In South Korea, the Industrial Safety and Health Law requires annual or biennial health screening examinations of all employees and many companies offer health check-ups as a welfare. Due to accessibility to health check ups, there are a lot of people who want to test it even though they are not the target of the screening. Daegu catholic university Hospital is one of the largest center in Daegu providing these health screening exams. Therefore, most of them were heathy people and the subjects were who underwent screening colonoscopy and carotid ultrasonography on the same day for annual or biennial health check up.

2) I am unfamiliar with carotid sonography being used as a screening test. On what basis was this introduced? Has it been evaluated in terms of reducing the morbidity and mortality from cerebrovascular disease?  What is its sensitivity and specificity when used as a screening test?

Answer) Thank you for your comments. We have added the comments for the article and added them as references.

According to our reference, CIMT is widely studied as a surrogate marker for detecting subclinical atherosclerosis for risk prediction and disease progress to guide medical intervention. However, there is no standardized CIMT measurement methodology in clinical studies resulting in inconsistent findings, thereby undermining the clinical value of CIMT. Increasing evidences show that CIMT alone has weak predictive value for CVD while CIMT including plaque presence consistently improves the predictive power. The study of sensitivity and specificity of carotid ultrasound was conducted. The results were shown that the threshold of peak systolic velocity >/=130 cm/s is associated with sensitivity of 98% (95% confidence intervals [CI], 97% to 100%) and specificity of 88% (95% CI, 76% to 100%) in the identification of angiographic stenosis of >/=50%. For the diagnosis of angiographic stenosis of >/=70%, a peak systolic velocity >/=200 cm/s has a sensitivity of 90% (95% CI, 84% to 94%) and a specificity of 94% (95% CI, 88% to 97%).  

Round 2

Reviewer 1 Report

Thank you for your revisions. This is much improved. For clarity, please see the below bolded, minor comments.

4) Please indicate how many had CIMT or presence of carotid plaques.

Answer) It is shown in Table 2. The number of carotid atherosclerosis (either increased CIMT and/or presence of plaque) was 171/548 (31%). It is not written in the paper, but the number of only plaque of carotid sonograpy was 74/548 (13%).

***I was asking for the breakdown of CIMT and carotid plaques. I think it may be important for the reader to have this included in the paper in the results section.

) Did you evaluate statistical interaction between sex and carotid atherosclerosis? Even though the main effects were not significant, there may be effect modification here by sex.

Answer) Thank you for the comments.

We analyzed the the interaction between sex and carotid atherosclerosis. According to the result, there was no statistically significant interaction between sex and carotid atherosclerosis. (OR: 0.972, 95% CI: 0.335 to 2.819, p-value: 0.959)

***Please indicate in your methods how you evaluated statistical interaction and in your results that you found none and include OR, 95% CI. 

3. Results:

1) Table 1 headings are confusing. Non-adenomatous polyp sounds like they had some other kind of polyp. Please reword to indicate no polyp.

Answer) Thank you for the comments.

 An adenomatous polyp is defined as a polyp with adenomatous tissue, as confirmed by pathology. Non-adenomatous polyp includes the results of inflammatory polyp, hyperplastic polyp, etc. and no polyp. “Non-adenomatous polyp” doesn’t mean “no polyp”.

***So you are grouping "other" polyps and no polyps for non-adenomatous polyp? Please add your definition of non-adenomatous polyp to section 2.2.2 for clarity as well as footnote Table 1.

Author Response

4) Please indicate how many had CIMT or presence of carotid plaques.

Answer) It is shown in Table 2. The number of carotid atherosclerosis (either increased CIMT and/or presence of plaque) was 171/548 (31%). It is not written in the paper, but the number of only plaque of carotid sonograpy was 74/548 (13%).

***I was asking for the breakdown of CIMT and carotid plaques. I think it may be important for the reader to have this included in the paper in the results section. 

Answer) I revised in the document, Table 1 and Table 3.

Did you evaluate statistical interaction between sex and carotid atherosclerosis? Even though the main effects were not significant, there may be effect modification here by sex.

Answer) Thank you for the comments.

We analyzed the the interaction between sex and carotid atherosclerosis. According to the result, there was no statistically significant interaction between sex and carotid atherosclerosis. (OR: 0.972, 95% CI: 0.335 to 2.819, p-value: 0.959)

***Please indicate in your methods how you evaluated statistical interaction and in your results that you found none and include OR, 95% CI.  

Answer) I revised in section 2.3 and Result.

3. Results:

1) Table 1 headings are confusing. Non-adenomatous polyp sounds like they had some other kind of polyp. Please reword to indicate no polyp.

Answer) Thank you for the comments.

 An adenomatous polyp is defined as a polyp with adenomatous tissue, as confirmed by pathology. Non-adenomatous polyp includes the results of inflammatory polyp, hyperplastic polyp, etc. and no polyp. “Non-adenomatous polyp” doesn’t mean “no polyp”.

***So you are grouping "other" polyps and no polyps for non-adenomatous polyp? Please add your definition of non-adenomatous polyp to section 2.2.2 for clarity as well as footnote Table 1. 

Answer) I revised in the document, Table 1

Thank you for your comment!!

Reviewer 4 Report

None

Author Response

Thank you for your review.